# Improving together: better science writing through peer learning

M. A. Stiller-Reeve[1], C. Heuzé[2,3], W. T. Ball[4], R. H. White[5], G. Messori[6], K. van der Wiel[3], I. Medhaug[7], A. Eckes[3], A. O'Callaghan[3], M. J. Newland[3], S. Williams[8], M. Kasoar[8], H. E. Wittmeier[9], V. Kumer[10]

[1] Uni Research Climate, Bjerknes Centre for Climate Research, Bergen, Norway
[2] Department of Marine Sciences, University of Gothenburg, Gothenburg, Sweden
[3] Centre for Ocean and Atmospheric Sciences, School of Environmental Sciences, University of East Anglia, Norwich, UK
[4] PMOD/WRC, Dorfstrasse 33, 7260 Davos Dorf, Switzerland
[5] Joint Institute for the Study of the Atmosphere and Ocean, University of Washington, Seattle, USA
[6] Department of Meteorology, Stockholm University and Bolin Centre for Climate Research, Stockholm University, Stockholm, Sweden
[7] Institute for Atmospheric and Climate Science, Department of Environmental System Science, ETH Zurich, 8092 Zurich, Switzerland
[8] Department of Physics and Grantham Institute, Imperial College London, London, SW7 2AZ, United Kingdom
[9] Department of Earth Science, University of Bergen, Allégaten 41, N-5007 Bergen, Norway
[10] Geophysical Institute, University of Bergen, Bergen, Norway

*Correspondence to*: M. A. Stiller-Reeve (Mathew.reeve@uni.no)

**Abstract:** Science, in our case climate- and geo-science, is increasingly interdisciplinary. Scientists must therefore communicate across disciplinary boundaries. For this communication to be successful, scientists must write clearly and concisely, yet, the historically poor standard of scientific writing does not seem to be improving. Scientific writing must improve and the key to long-term improvement lies with the early-career scientist (ECS). Many interventions exist for an ECS to improve their writing, like style guides and courses. However, momentum is often difficult to maintain after these interventions are completed. Continuity is key to improving writing.

This paper introduces the ClimateSnack project, which aims to motivate ECS's to develop and continue to improve their writing and communication skills. The project adopts a peer learning framework where ECS's voluntarily form writing groups at different institutes around the world. The group members learn, discuss and improve their writing skills together.

Several ClimateSnack writing groups have formed. This paper examines why some of the groups have flourished and others have dissolved. We identify the challenges involved in making a writing group successful and effective, notably the leadership of self-organized groups, and both individual and institutional time management. Within some of the groups, peer learning clearly offered a powerful tool to improve writing as well as bringing other benefits, including improved general communication skills and increased confidence.

# 1 Introduction

Peer learning –within writing groups– offers a powerful tool to motivate early-career scientists (ECS's) to improve their writing and communication skills (Schultz, 2010; Colton and Surasinghe, 2014). In this paper, we review the ClimateSnack project: a peer learning framework of self-organized writing groups for ECS's. The project was named ClimateSnack because the members would write blog posts (or "snacks") about climate-related topics. The members would collaborate with the rest of their writing group to improve their "snacks" before publishing online. Recently the project has been renamed SciSnack and welcomes ECS's from all scientific disciplines. In this paper, we detail the successes and challenges of the project so far and the lessons learnt, with the view that these lessons may inform similar projects.

Communication with scientists within one's own discipline is a vital aspect of a scientist's work. Additionally, as science becomes increasingly interdisciplinary (Porter and Rafols, 2009), the need also increases for scientists to clearly communicate across disciplinary boundaries (Aboelela et al., 2007; Langdon-Neuner, 2009). This is particularly important in the realms of climate change and natural hazards (Mostert and Raadgever, 2008; Donnelly, 2008; Alexander, 2007), where solutions inherently require interdisciplinary approaches with close links to policy. Until now, scientists have primarily communicated in written form, with the scientific article as the main medium.

Unfortunately, scientific articles often alienate their intended readers (Halliday, 1989), due to a verbose and opaque writing style (Wilson, 1952; Trelease, 1958; Pinker, 2014). Some of the fundamental weaknesses of scientific writing stem from issues of reader expectations and information placement (Gopen and Swan, 1990). Complaints have circulated for decades about the standard of scientific writing. In 1952, Wilson asked the following of the scientific writer:

"Does the writer really want to convey information to his readers, or is he trying to impress them with his own genius? Unfortunately, some scientists suffer from an inferiority complex, which continually compels them to bolster their egos by writing papers so obscure that even the most brilliant specialists in the same field cannot understand them. What a triumph!" (Wilson, 1952) (p.357)

Over half a century later, Wilson's complaint still resonates (Langdon-Neuner, 2009; Heatwole, 2008). Indeed, the situation may not have improved at all (Wells, 2004; Geerts, 1999; Besley and Tanner, 2011). O'Donnell (2000) humorously suggests that the reader of a scientific article has two choices:

"do the writers' work for them by trying to work out what they are trying to say, or throw the journal aside and set about doing something less demanding like quarrying granite."

The lack of improvement in scientific writing may be attributed to ECS's learning writing from the vices of their seniors: poor writing breeds poor writing (McCartney, 1955; Schultz, 2010). One option to break from this vicious cycle is for ECS's to receive training and gain control of their own writing. They must learn to better consider their audience, and communicate their science more clearly and concisely. A welcome side effect of this improvement process is that writing may also become more enjoyable!

Several methods to improve writing already exist. There are many excellent style guides on the market to help ECS's improve individually (Greene, 2013; Schimel, 2012; Montgomery, 2003; Schultz, 2013). Online writing courses allow ECS's to watch online lectures, write assignments and peer-review fellow students. In addition, many universities offer graduate training courses in writing and communication, and indeed there is a call for writing to be integrated into mainstream science education (Brownell et al., 2013; Boice, 1990). One notable example showed that when writing skills were integrated into hydrology courses, the students improved both their writing and critical thinking skills (Carlson, 2007). Several other studies have found that students benefit immensely from this type of training (e.g. (Motavalli et al., 2003; Kokkala and Gessell, 2002; Holyoak, 1998; Woodford, 1968; Bean, 2011). One problem with such training options is that they are often short-term. Once the courses end, motivation can quickly ebb, leaving the student to work alone to improve further. The other problem is that these training options are not open to all interested scientists. However, solutions are available that are free to everyone.

Online writing courses are often freely available and the development of the ClimateSnack project was inspired by one such course lead by Kristin Sainani at Stanford University (http://online.stanford.edu/course/writing-in-the-sciences). During the course, students peer-reviewed each other's writing and an active online discussion community flourished. However, when the course ended, the community dissolved. The founders of ClimateSnack wanted to duplicate this interactive forum and also encourage continuity. Therefore, in-situ writing groups were proposed. Discussion would then continue within the groups and between international participants via the project's website.

Writing groups offer a flexible approach that can be implemented both in connection with university science courses (Ferguson, 2009) or in a self-organized fashion by students (Maher et al., 2008; Wegener et al., 2014). The successes of writing groups at the doctoral level are well documented (Aitchison, 2009), including in interdisciplinary settings (Colton and Surasinghe, 2014; Kokkala and Gessell, 2002). The benefits include: improved writing and communication (Grant et al., 2010), improved critical thinking (Bean, 2011; Maher et al., 2008), improved support networks (Grant et al., 2010), and increased confidence (Ferguson, 2009). Such benefits derive primarily from face-to-face peer feedback and the continuity within the groups (Caffarella and Barnett, 2000), which other writing interventions may not offer so readily. These writing groups provide safe, non-judgemental environments for ECS's to practice, make mistakes and improve, which may be particularly pertinent for non-English speaking researchers. Despite some writing group mishaps (Nairn et al., 2014), the majority of literature agrees that writing groups offer the "winning formula" (Grant et al., 2010) for ECS's who aim to improve their writing.

The objective of the ClimateSnack project was to encourage ECS's to self-organize writing groups to improve their *basic* writing skills, and thereby also their scientific writing skills. These basic writing skills alone are not sufficient to write quality scientific articles. However, these sills they are important ingredients for overall improvement. In the groups, the participants would write short articles about their science or topic of interest. The audience for these snacks are fellow ECS's. In other words, the audience is assumed to be scientifically literate but not from a single research discipline. The participants therefore also gain experience in interdisciplinary communication, where audiences will have different levels of

familiarity with the ideas and themes being discussed. Once an author had written an article, the rest of the group provides constructive feedback, which the author uses to improve the text. The author publishes the finished article on the project's website (www.climatesnack.com). The web page also acts as a space for the participants to network with like-minded researchers from around the world. ClimateSnack has two unique elements: it is self-organized by ECS's and it tries to build an international community around writing skills in science.

In the next section, we introduce the learning process around which ClimateSnack was built. The successes and challenges that the different groups encountered will be presented in section 3. Results from an informal questionnaire answered by one group are presented to illustrate how the participants benefitted. Besides this, we did not evaluate ClimateSnack's impact using quantitative metrics from the beginning. This paper therefore takes a narrative approach and reports on the experiences of the members and their groups. These narratives are provided by the present paper's authors, all of whom have been involved in the project development. Some authors founded and managed ClimateSnack, whilst other authors started groups that are either still running or have since dissolved. All the authors have been active participants and have written and posted articles on the ClimateSnack website. We began to collate these experiences in a group meeting at the European Geosciences Union General Assembly in Vienna in 2015. Further discussion has taken place via email and video conferencing. In section 4, we generalize the lessons learnt and conclude with some thoughts about the future.

## 2. The process

The ClimateSnack founders designed a writing process, which is still used as a guide for writing group meetings. ClimateSnack participants write short texts about their own research or a scientific topic of interest. These texts were dubbed as "snacks" and are usually 400 to 700 words long.  Snacks are meant as snapshots of the authors' interests, rather than full-length texts suitable for peer-reviewed journals. The snacks develop from an initial writing idea to a finished product through an iterative process of individual writing and group feedback. We now describe this process, as represented in corresponding Fig. 1 by an author who is preparing her first snack:

1. **Ideas (Meeting 1)**

   Before the author can write, she needs a topic. The group listens to the author's ideas and gives feedback. This discussion focuses on helping the author find the main question(s) that her snack will answer, and how the snack might be structured.

2. **Write a structure**

   Before the next meeting, the author writes a structure for her snack. This process entails specifying the main question, and ideas for each paragraph. The next step is to write the topic and stress sentences for each paragraph so that the flow of ideas is clear.

3. **Feedback (Meeting 2)**

At the next meeting, the author presents this structure to the group. The group will give feedback on the flow of ideas and whether the ideas actually contribute to the main aim of the article. At this stage, the author will likely have to delete (or add) paragraphs to improve clarity.

**4. Write a draft**

The author writes a draft snack with the improved structure and adds relevant figures. The draft should include references where necessary with proper citations. This draft is distributed to the other members of the group prior to the next meeting.

**5. Feedback (Meeting 3)**

Before the meeting, the other group members should have read the author's snack and noted down feedback directly on to the snack itself. These notes are passed to the author at the end of the meeting. We found that this encouraged meeting attendance. However, groups could also consider other feedback methods, like online editing software (e.g. Google docs). At the meeting, the author reads her snack aloud. Reading aloud has been shown to help develop writing skills (Gibson, 2008), and quickly pinpoints sentences and sections that need improvement. After the recital, the feedback discussion commences. For this discussion to work well, we need structure, knowledge and a good dose of courtesy.

The ClimateSnack founders encouraged the discussion to be structured around the "funnel model" as illustrated in Figure 2. This model helps to guide the discussion from the big issues that irritate the reader the most, such as paragraph structure (Hofmann, 2014), to smaller issues, like spelling and grammatical mistakes. The background knowledge for the discussion may come from previously read books or previously completed courses. If this knowledge is difficult to come by, the ClimateSnack website freely provides an entire online writing course, advice from experts (see Figure 3), as well as book reviews from some of the ClimateSnack members. In this way, ClimateSnack targets the deficiencies in academic writing, and promotes the advice of experts (e.g. Gopen and Swan, 1990; Somerville and Hassol, 2011). In addition, the discussion needs to be conducted with courtesy and humility. A simple rule is applied: Only say something that you could accept hearing yourself.

**6. Improve and publish online**

Once the author has used the peer feedback and notes to improve her snack, she is ready to publish online. She uploads her text and figures to the ClimateSnack website. Figure 3 includes some screenshots from the website, including the homepage and an example snack.

Once the snack is published the writing process can start again, resulting in a continuous process and hence continuous improvement. The process is also flexible, and group leaders have the freedom to adapt it as they see fit.

Most group meetings lasted between 1-2 hours once every 3-4 weeks. Each completed snack was discussed for 20-30 minutes depending on how many snacks were under consideration, and at what stage they were at. Some time was also

often left at the end of the group meetings for general discussions and brainstorming. Some group meetings had up to 20 participants, but usually 5-10 people attended the meeting in the groups that regularly convened. The group leader was in charge of guiding the discussion and following the framework illustrated in Figure 2.

The writing group process can lead to numerous other benefits. For example, participants network amongst themselves and learn about each other's research. The website promotes discussion and networking on a global scale. Publishing on the website also gives members experience with different media and outreach opportunities. The group leaders gain a valuable set of transferable skills by organizing the meetings and moderating the feedback sessions.

ClimateSnack writing groups started in several places around the world. In some places, the groups worked extremely well, and in others, they lasted a fleeting moment. In the following section we consider case studies of a successful group, and of groups that lost momentum.

## 3. Results

The ClimateSnack project started at the University of Bergen, Norway in January 2012. Since then, ClimateSnack writing groups have produced over 100 snacks by 49 members. In total, 10 writing groups started (all within Europe and North America), of which 3 are still active (one in Norwich, UK and two in Bergen, Norway). It is clear that the majority of groups encountered difficulties. We can learn important lessons by comparing what caused some groups to flourish and others to dissolve.

### 3.1 Group success

In 2013, a small pilot group started at the University of East Anglia (UEA). In the first two years, this small group developed into a successful writing group that has published 25 snacks by 11 authors with two collaborative posts by the whole group. Members of this group have identified three key aspects that they believe have contributed to the group's success: the social atmosphere, high attendance with gradual initial growth, and strong leadership.

The UEA writing group places a strong emphasis on the social atmosphere of the group meetings. The resulting friendly ambiance is thought to facilitate the high attendance rates. Group members also share a common desire for communicating their science. Although each meeting has an agenda, off-topic conversations are tolerated and have led to new ideas for future posts. The social atmosphere further encourages members to provide honest and constructive feedback, but also to ask for help or advice if needed.

Another key attribute to the writing group's success is the high attendance rates. At present, there are 21 members, of whom 15 are active and regularly attend meetings. The monthly meetings are arranged to take place immediately after the department coffee break, which may help maintain high attendance rates. These large numbers decrease pressure on individual members to contribute. Over time, this decreased pressure could obviously be counter-productive. However, it also allows new members to only observe during their first meetings, and contribute with their own writing when they feel

comfortable. In addition, a sufficiently large group means that, if not all members can attend every meeting, the group is still large enough to function, and the peer learning process can continue. During the pilot phase of this writing group the size was limited to 5 members. Since then, the group size has steadily increased. New members benefit from the experience that has developed within the group.

Members of the UEA group have described the leadership as "strong, but friendly", and suggest that this may play a key role in sustaining the large, enthusiastic, and productive group. The leaders have played an active role in raising attendance by introducing new members to the group, and have also set an example by writing multiple posts themselves. The members feel that there needs to be a balance when encouraging people to write. On one hand, a leader can gently inspire people to write. On the other hand, a leader might insist too much and scare people away.  Recently, the leadership role of this group

has been passed on to newer group members; the group remains keen to continue the collaborative learning process that has been successful so far.

Of course, the true success of the project depends on whether participants have improved their writing and communication skills. Since ClimateSnack is self-organized and inherently lacks any formal assessment process, we have not attempted to rigorously measure this outcome. However, an informal survey amongst 16 active UEA group members

shows that all of them believe that ClimateSnack has helped to improve their writing style. That includes members that have published blog posts and members that have not yet published. The unpublished participants felt that they have benefitted from taking part in the discussions. Furthermore, 12 of the 16 respondents felt that their confidence has increased when presenting their research to both the scientific community and the general public.

Anecdotal evidence emerged about some individual successes within the ClimateSnack community. For example, one of

the present authors used the lessons learnt to improve a research paper and get it published in a peer-review journal. A previous submission of the paper had received comments like, "the excessive use of passive voice makes it difficult to understand and quite dry". The final accepted version of the paper received much better feedback, with one reviewer stating that, "this is a well written paper". ClimateSnack also helped with other outreach channels. A member of one group got her snack translated into Norwegian and published in one of the biggest newspapers in Norway.

**3.2 Group dissolution**

Unfortunately, not all of the writing groups have achieved long-lasting success. Several common factors surfaced when we discussed why groups dissolved. Some factors were related to large-scale, institutional issues, and others were related to the dynamics within specific groups. Here, we offer an overview of some of these issues, as a resource to future ClimateSnack development and other similar peer learning projects.

Writing groups will simply not function without motivated participants. Unfortunately, research has shown that humans perceive future rewards as less valuable than immediate rewards (Green et al., 1994). ClimateSnack offers rewards, but they are neither immediate nor directly quantifiable. Writing groups ideally provide a gradual but continuous honing of one's written communication skills, but there is no set time horizon for this process. Without a set time horizon other

commitments quickly prevail and motivation falls. For an ECS, this means the struggle to juggle writing group participation with teaching duties, seminar attendance, research cruises, study exchanges, and their own research. Contributing to an entirely voluntary project like ClimateSnack is inevitably viewed as something that can be sacrificed in favor of other research commitments. At one institution the ClimateSnack group faced competition with a new Doctoral Training Centre that introduced a range of structured professional development courses. The ClimateSnack project clearly needed other motivational attractions rather than just writing improvement.

The project website (www.climatesnack.com) was intended to motivate ECS's to participate in the project. The website provided a common platform for publication, but more importantly for discussion and networking between the participants. This network would hopefully also motivate groups to continue to write. However, in most groups, this networking was never appreciated as an important component in the writing process. Isolation and relatively little contact between groups mean that group development is very dependent on the group leadership and internal dynamics.

The very concept of ClimateSnack as a peer learning project means that the group leaders are often learning too. Some groups initially relied heavily on a few highly motivated individuals, sometimes with prior leadership experience. When these individuals stepped down, the groups were often left fragile. This fragility may have been enhanced by the lack of clear learning structure in the groups.

This lack of clear structure and objectives sometimes caused confusion and impacted the internal workings of some groups. Even though the ClimateSnack founders suggested a writing process and discussion structure, some groups had sterile feedback sessions focusing solely on grammatical subtleties. It was clearly not enough just to inform the group leaders about the suggested process for writing and the funnel model for discussions. In hindsight, the website lacked the resources needed to help the group leaders develop their groups.

Group development was sometimes hindered by internal dynamics. In groups that attracted like-minded scientists who already had a strong focus on science communication, a situation of "preaching to the converted" arose. Feedback suggests this may have intimidated prospective members who were less experienced in the field. Conversely, some members have perceived a general lack of science communication knowledge in their group, resulting in a "blind leading the blind" scenario.

Some groups experienced other issues specific to their institute. For example, one group possibly had too much initial success, with up to 20 people meeting for the first meetings. At this early stage of the group's development, the large number of members impacted negatively on the feedback and discussion process, and presented logistical challenges such as booking suitable rooms. These difficulties are thought to have discouraged some members from attending successive meetings. Attendance continued to dwindle and the group disbanded.

**4. Discussion and Conclusions**

ClimateSnack's main objective was and still is to help ECS's improve their basic writing skills in order to improve scientific articles and other types of communication. ClimateSnack fulfils this objective by providing a continuous and free framework of peer learning. The peer learning occurs through interaction within self-organized writing groups established around the world. Participants write, discuss and improve together.

No matter how long the different writing groups lasted, they brought ECS's together and created a forum for discussing writing skills for various goals such as grant proposals, scientific articles and conference abstracts. Friendships and community have been built around ClimateSnack (the present author group being an example) allowing ECS's to seek out advice and feedback even if their groups discontinued. Despite the majority of ClimateSnack groups dissolving, some have done very well and thrived. We have presented a case study of one of the groups that succeeded and can summarize the main perceived reasons for group success as follows:

- Strong social aspect
- High attendance rates
- Gradual development
- Strong leadership

The gradual development of the group at UEA was seen as key to their success. The group grew, as the leaders felt more confident. The group has now grown to an optimal level, which the leaders feel comfortable managing. The meetings are well attended, which gives newcomers the opportunity to observe and contribute when they feel ready. In contrast, another group came into difficulties when too many people came to the meetings and the leaders had difficulties organizing the discussion. In this scenario, one may contemplate organizing several writing groups at one location. This solution depends on whether enough people are available to lead multiple groups.

We observed that in the groups that thrived, several other positive outcomes resulted from the peer learning approach. Participants reported that both their writing skills and confidence had improved. This was reflected in the discussions we had and the short, informal survey that the UEA group carried out with its members. The group leaders have also learnt important organizational and chairing skills, all of which can be transferred to project and research leadership positions in the future.

The main reasons for group dissolution that our discussion emphasized are:

- Low motivation: perceived as a waste of time
- Poor group discussion dynamics
- Too much pressure on group leaders

- No handover of leadership
- Competition from other commitments
- Weak international network
- Unclear objectives

We suggest that to increase the chances of group success, learning resources online should better advise participants and group leaders. This advice could include clearer information about the objectives of the ClimateSnack project, so that participants better understand the long-term benefits of taking part. The online advice should also include tips about how to start a group, how to run a group, how to guide discussions, and how to hand over leadership. This may help to reduce the

pressure some of the group leaders felt. The website must also be further developed to better encourage international networking and communication between writing groups. If funding is not available for such developments in this project or others, then social media platforms should be used more actively.

The motivation and time-related problems that some groups encountered can be reduced by effective co-leadership. In this way, when one leader is away or departs then the group can still continue to function. This approach also reduces the

pressure on individual leaders and provides a more social and interactive experience.

Competition from other research activities appeared to hinder some groups from continuing; if these institutionalized activities are concerned with improving science communication, then this is a positive development in training opportunities for ECS's. If these activities are effective, then the need for a ClimateSnack writing group is diminished anyhow.

The ClimateSnack project had ambitious objectives to unite ECS's across the world to improve their writing skills together. This was born out of a continued need to improve our scientific article writing and also other types of science communication. Even though we did not evaluate the project using continuous quantitative metrics, this paper is based on the honest accounts and narratives of many of the ClimateSnack participants who make up the paper's author group. We would recommend that similar projects consider tracking evaluation metrics from the beginning. Through our collected experiences

we have seen that ClimateSnack faced several challenges, but the successes show that peer learning through self-organized writing groups can be a valuable approach to achieve improved writing in science. With a greater understanding of why some groups did not flourish, we can improve future initiatives. It is true that the international network did not develop in the way the ClimateSnack founders first envisaged. However, within successful writing groups, solid support networks arose and many members discovered that writing could be a pleasurable activity. This pleasure can be transferred from writing

snacks to writing scientific articles for a specialized audience. Even though the style is different, basic writing skills still apply. Hopefully one day, we may all write scientific articles that are enjoyable to write *and* to read, whilst also moving science forward.

**Author contributions:**

M. A. Stiller-Reeve founded developed the ClimateSnack project and managed the paper writing process.

C. Heuzé founded a ClimateSnack writing group, co-directs the ClimateSnack project and contributed with writing.

W. T. Ball co-founded the ClimateSnack project and contributed with writing.

5    R. H. White co-ordinated ClimateSnack events and contributed with writing.

G. Messori founded a ClimateSnack writing group and contributed with writing.

K. van der Wiel has been an active ClimateSnack participant and contributed with writing.

I. Medhaug has been an active ClimateSnack participant and contributed with writing.

A. Eckes has been an active ClimateSnack participant and contributed with writing.

10    A. O'Callaghan founded a ClimateSnack writing group and contributed with writing.

M. J. Newland founded a ClimateSnack writing group and contributed with writing.

S. Williams led a ClimateSnack writing group and contributed with writing.

M. Kasoar led a ClimateSnack writing group and contributed with writing.

H. E. Wittmeier has been an active ClimateSnack participant and contributed with writing.

15    V. Kumer has been an active ClimateSnack participant and contributed with writing.

**Acknowledgements:**

The ClimateSnack project has been funded through generous contributions from Uni Research AS, Bjerknes Centre for Climate Research (via The Christie Conference), The Norwegian School for Climate Dynamics, The University of Bergen, The Grantham Institute for Climate Change, and The Lions Club. The figures were produced by Thorbjørn Kongshavn at Kongshavn Design (www.kongshavndesign.no). We thank all the ClimateSnack participants over the past couple of years for their efforts in developing the project. We thank the two reviewers, Heather Galindo and Iain Stewart, for their constructive comments that greatly improved this paper.

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

**Figure 1.** The ClimateSnack writing process

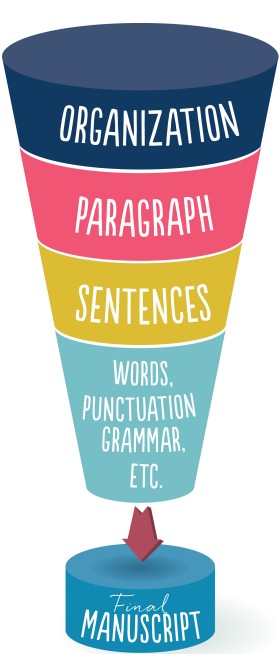

**Figure 2**. The writing/editing Funnel Model used to guide feedback discussions in the ClimateSnack writing groups, based on Schultz (2013) and Snellman (1982).

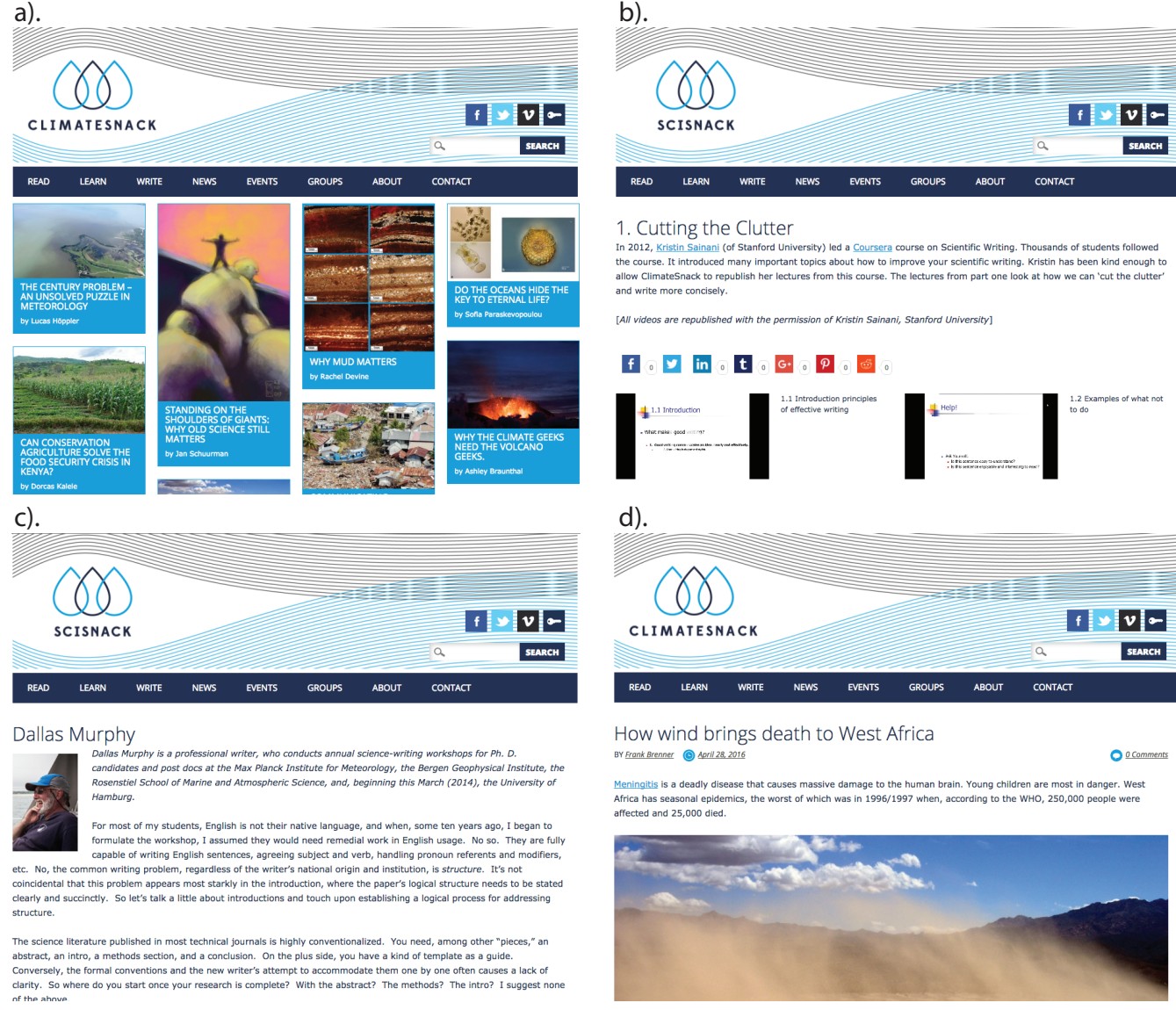

**Figure 3:** Screenshots from the ClimateSnack/SciSnack website showing, a). the homepage, b). one of the chapters from the online writing course from Kristin Sainani, Stanford University, c). an expert advice from author Dallas Murphy, and d). an example snack.