# Peer review of "Improving together: better science writing through peer learning"

_Hydrology and Earth System Sciences, 2016_

## Referee Comment (RC1) · H.M. Galindo (Referee) · 24 Feb 2016

GENERAL COMMENTS: Although this is not a conventional research paper, the topic of effective communication (writing, in particular) is essential to success in any scientific field. I think this paper serves two important functions: 1) Providing an introduction to the research-based evidence that exists about effective writing, and 2) Presenting a method for creating and sustaining a peer-facilitated writing group for early career scientists. I do think this paper could be strengthened by the inclusion of metrics for the effects the writing groups had on the participants, so perhaps developing such an assessment could be a future research goal for the authors?

SPECIFIC COMMENTS: In no particular order, here are a few additional thoughts: - In lieu of a formal assessment, are there other data that could be used to support the

claim that these writing groups are beneficial? I'm thinking of things like acceptance rates of papers, grants awarded, or similar metrics for the participants during the time they were involved in these groups. I realize that the timeframe (since 2012) makes this difficult, but even a few qualitative examples could be useful. - I like the explicit detail provided about the writing process and the accompanying figure. I was wondering if only one member of the group is working on a piece at a time? Also, how long are these meetings? With 20 people, is each one giving a few minutes of feedback or is it more of a free-for-all discussion? - Do the groups use any online co-editing software (I'm thinking of something like Google docs) to share comments or are they all hand-written on printed copies? - On page 3 line 21, you seem to imply that improving basic writing skills will automatically translate into improved scientific writing skills. I think the former is necessary, but not sufficient, for the latter. Can you be a bit more explicit about some of the skills you do/do not think are covered by this process? I'm also thinking about how skills related to creating effective blogposts do/do not relate to other types of writing required by scientists (e.g. see next point). - This is more about the concept than the paper, but have you thought about using these groups to provide peer-review for other types of writing ECS's are faced with? I'm thinking about things like grant proposals, scientific papers, abstracts for conferences, etc. This might entice ECS's who aren't committed to writing blogposts, but would engage in activities more focused on something they already have to do. - Along the same lines, for groups not comfortable with how to give feedback, I wonder if a structured rubric would be a good complement to the process depicted in Figure 2? I've been using rubrics based on the goals of the writing product in my undergraduate scientific writing class and it seems to help the students get started on first assessing the content (function) and then figuring out how the structure (form) could best support the ideas.

TECHNICAL CORRECTIONS: A few small things I noticed: - Page 2, Line 11: I think a word is missing between "communicate" and "disciplinary" - There seems to be incon-sistency in whether or not the first line of a new paragraph is indented (e.g. page 6 line 22).

---

## Editor Comment (EC1) · 3 Apr 2016

Improving together: better science writing through peer learning

Review: I Stewart

The paper is a descriptive account of the development of academic writing groups aimed at improving basic scientific writing skills for early career scientists working in climate science / geoscience. Although the paper does not attempt to provide a critical or quantitative appraisal of the effectiveness of the intiative, it is a useful contribution, sketching out a justification for improving academic writing and charting first-hand experiences of writing groups tied to the ClimateSnack online blogpost initiative. In that regard it is a potentially valuable account of an emerging approach that is likely to be of interest to the readership of this particular special issue. I certainly found myself trying

to imagine setting something like this, and so would appreciate a bit more clarity and detail on a number of fundamental points that I feel the authors have rather glossed over. Indeed, one is the actual origins of the ClimateSnack project tself, which is introduced rather unobtrusively (line 20, page 3) but I think a sentence or two about its background context would help the reader. The substantive points are outlined below:

Firstly, although the authors state that scientists '…must learn to better consider their audience, and communicate their science more clearly', for me, the paper wasn't especially clear on who precisly the ECRs are writing for. The paper implies the audience is both scientists within their own discipline and those across disciplinary boundaries,' so presumably the focus remains squarely on academic writing rather than drifting into popular science writing (for which there is a far richer science communication literature that is not called on here). I appreciate that there is a continuum of writing styles that can be invoked fo reach different audiences, but it would be helpful if the article could add a sentence making clear the specific readership that ClimateSnack participants are targetting, as that sets the rubric for ll that follows in terms of how they prepare and hone their contributions.

Secondly, and in a similar vein, although the scheme seeks to improve ' 'basic writing skills, and thereby also their scientific writing skills' it is never made explicit what are the deficiences that the initiative is trtying to redress. As the paper notes, there are plenty of academic voices bemoaning the quality of academic writing but precious few that actually dissect the problem in a meangingful way; one telling exception is Goben, G.D. & Swan, J.A. 1990. The Science of Scientific Writing. American Scientist, 78 (Nov/Dec), 550-558. While the authors direct the reader to papers that presumably shed light on the substance of this problem, that is not especially helpful for an inidividual interested in improving their writing. Given that this paper attempts to set out the theoretical basis for this practice, it is important to be as explicit and transparent about how those championing the ClimateSnack initiative preceive the fundamental weaknesses and limitations in mainstream academic writing. A short section or paragraph
on this should be added.

Thirdly, the reason that the basic deficiencies need to be made more explicit is that it is not immediately clear how - or indeed, if - the ClimateSnack initiative is addressing the core communication issues raised by practitioners working in climate science arena. This is alt is an arena that is pretty frequently addressed by those publishing in science communication. One prominent contributer is Richard Somerville, prof at Scripps and the science director of the nonprofit project 'Climate Communication', who has written and blogged extensively on this and highlights a range of issues that do not appear to feature in the ClimateSnack developmental process. For example, his review in Physics Today (Somerville, R.C.J. & Hassol, S.J. 2011. Communicating the science of climate change. Physics Today, October, 48-53.) critiques the conventional academic model of writing and presents some clear recommendations for making climate science writing more accessible. It may be that the authors would disagree with his contentions, but the point is that it is impossible to tell because there is no indication of to what extent the now pretty extensive critical literature on climate science communication is infusing and informing the ClimateSnack initiative. To put it bluntly, is ClmateSnack simply a self-help support group for a particular scientific cohort or is it actively carrying forward the experience of climate science communicators? If it is the former then OK but that more limited remit needs to be stated; if it is the latter then the paper needs to be far more explicit on how participants are building on what is out there.

Fourthly, could some indicative content from the site be included? It could be a screenshot or two , or brief excerpts from posted articles. I found myself frustrated that all I was reading about what the process and could not view the product (at least, not without accessing the webpage - perhaps a deliberate ploy). I appreciate that it is a sensitive issue, but there must be exampes of 'good practice' that the team feel showcases what the ClimateSnack initiative can achieve in recasting academic writing.

Finally, many readers of this paper will lament the omission of some kind of empirical analysis of its efficacy. The informal 'survey' of why groups succeed or not simply adds

to the frustration of not getting a better sense of how effective this novel approach is; to make a useful contribution more specifics on what this survey involved should be given. Overall, I'm sympathetic to the nascent nature of the initiative and also to the difficulties in determining meaningful metrics, but I agree with the other reviewer that there are indicative measures that the authors could and should consider regarding readership and impact. On a related note, perhaps the authors could mention something more about the international community that has been fostered as a result of ClimateSnack?

In summary, the paper is an enthusiastic but rather uncritical account of one initative to counter perceived limitations in our current academic writing provision. While I share many of the authors' basic contentions and find the ClimateSnack an intriguing and welcome development, the paper as it stands lacks substance in key areas and I would ask the authors to attempt to address these in their revisions.

––––––––––––––––––––––––––––––––––

---

## Author Response (AR1)

**Response to Reviewer 1 RC1 including the applied changes (June 2016)**
**hess-2016-13**
**Improving together: better science writing through peer learning**

**Reviewer quote:** **I do think this paper could be strengthened by the inclusion of metrics for the effects the writing groups had on the participants, so perhaps developing such an assessment could be a future research goal for the authors?**

**Reply:** Thanks to the reviewer for a very constructive suggestion. Since we received both the reviews we have had an intensive discussion within the author group about such metrics.

The author group agrees that getting formal metrics in retrospect would not be desirable. The metrics from the UEA group are very clearly described as "informal" and we only use these as indications of the effects.

Metrics are something the project managers should absolutely have considered at the beginning of the project. However, ClimateSnack has always been a voluntary project where many of us have used our free time, with little or no funding, to develop groups, support authors, and write ourselves. We feel that formal metrics would have taken considerable time to develop and instigate. This would have required considerably more funding.

The reason we think that this would have been more complicated than maybe first imagined, is that the effects of such writing groups are so multi-faceted. As we have discussed in the paper, it's not just about writing quality; the effects are also concerned with general confidence, critical thinking, and network building. We must also consider the writing *process* in addition to the quality of the final product.

We also discussed how we could have measured improvement in writing quality. This would likely have been left up to the participant to judge himself. One of our authors pointed out a substantial challenge with this. He was a very confident writer before he joined ClimateSnack. However, through the writing process and group feedback, he started to understand that his writing was not as skillful as he first assumed. If he had filled out a self-assessment form before and after his participation, he may have actually perceived a decrease in writing quality, whereas objectively his writing had actually improved.

Moreover, ClimateSnack is an initiative where virtually all participants are early-career researchers. Most objective metrics would require members to have relatively long control periods both before and after joining ClimateSnack. The former requirement already excludes the large majority of members, who joined ClimateSnack during their Ph.D.

As part of the review process we carried out a survey to gather information such as acceptance rates of paper and abstracts, success in applying to travel awards etc. However, we quickly realised that most of our members joined ClimateSnack very early during the career, and that the changes in the metrics perhaps reflect more the natural development of their scientific abilities than the benefits of our writing groups.

Our most important point is that we feel our whole paper is already a metric. Indeed, it is not a quantitative metric (as alluded to by the reviewers), but it is a narrative metric. We feel that this is both more valuable and robust than an *ex post* survey, which would encounter all of the issues described above. The whole paper is built upon the narratives of 13 of the most active ClimateSnack members and others. Everyone in the author group has been a member of a ClimateSnack writing group. Some started groups that succeeded, whilst others started groups that dissolved. All the authors have built a network internationally (case in point, the present paper), and also extended their networks where they work.

Proposed action: We will add text explaining that we take a narrative approach in this paper and emphasizing how much the authors have contributed to this project. We will also add text to say that the lack of quantifiable metrics may be a limitation, but that this is something we could consider for the future. Similar projects should certainly consider metrics from the beginning, if funding allows it.

Action taken: We had to change around the end of the introduction slightly to accommodate these changes. The last paragraph of the introduction now reads:

"*In the next section, we introduce the learning process around which ClimateSnack was built. The successes and challenges that the different groups encountered will be presented in section 3. Results from an informal questionnaire answered by one group are presented to illustrate how the participants benefitted. Besides this, we did not evaluate ClimateSnack's impact using quantitative metrics from the beginning. This paper therefore takes a narrative approach and reports on the experiences of the members and their groups. These narratives are provided by the present paper's authors, all of whom have been involved in the project development. Some authors founded and managed ClimateSnack, whilst other authors started groups that are either still running or have since dissolved. All the authors have been active participants and have written and posted articles on the ClimateSnack website. We began to collate our experiences in a group meeting at the European Geosciences Union General Assembly in Vienna in 2015. Further discussion has taken place via email and video conferencing. In section 4, we generalize the lessons learnt and conclude with some thoughts about the future.*"

We also included the following in the conclusion, in order to explain the possible limitation of the lack of quantitative evaluation metrics:

"*Even though we did not evaluate the project using continuous quantitative metrics, this paper is based on the honest accounts and narratives of many of the ClimateSnack participants who make up the paper's author group. We would recommend that similar projects consider tracking evaluation metrics from the beginning. Through our collected experiences we have seen that ClimateSnack faced several challenges, but the successes show that peer learning through writing groups can be a valuable approach to achieve improved writing in science.*"

**Reviewer quote: In lieu of a formal assessment, are there other data that could be used to support the claim that these writing groups are beneficial? I'm thinking of things like acceptance rates of papers, grants awarded, or similar metrics for the**

**participants during the time they were involved in these groups. I realize that the timeframe (since 2012) makes this difficult, but even a few qualitative examples could be useful.**

**Reply:** The reviewer is presenting some nice ideas for future assessments. As part of the present review process we attempted to carry out a survey to gather such information amount previous participants. However, we quickly realized that most of our members joined ClimateSnack very early during the career, and many had not submitted grant proposals or papers before joining. It was therefore difficult to judge objectively if any improvement had been made. Again we fall back on our argument that the narrative metrics (stories) are the most appropriate way to convey the results of ClimateSnack, without carrying out a professionally-designed survey.

Some qualitative examples could be useful as the reviewer says.

One example comes from the ClimateSnack founder. His writing went from being "heavy and passive" by one reviewer, to "Excellent" by another reviewer just two years later. He and another co-author also organized a successful writing workshop course in Uganda in 2015 where the participants worked together in small groups to improve their writing following a series of short lectures. Neither of these developments would have happened if it wasn't for the time invested in ClimateSnack.

Also, another participant had one of her snacks published online in one of the biggest newspapers in Norway. Indeed, someone else had to translate it into Norwegian, but the story and the flow were the same.

Proposed action: Since we are concentrating on narratives evidence, we can include some of these anecdotes, if the reviewer agrees.

Action taken: We have included a couple of short anecdotes in the results in section 3. The following text was added:

"*Anecdotal evidence emerged about some individual successes within the ClimateSnack community. For example, one of the present authors used the lessons learnt to improve a research paper and get it published in a peer review journal. A previous submission of the paper had received comments like, "the excessive use of passive voice makes it difficult to understand and quite dry". The final accepted version of the paper received much better feedback, with one reviewer stating that, "this is a well written paper". ClimateSnack also helped with other outreach channels. A member of one group got her snack translated into Norwegian and published in one of the biggest newspapers in Norway.*"

**Reviewer quote: I like the explicit detail provided about the writing process and the accompanying figure. I was wondering if only one member of the group is working on a piece at a time? Also, how long are these meetings? With 20 people, is each one giving a few minutes of feedback or is it more of a free-for-all discussion?**

**Reply:** We can certainly include more specific information about the meetings in section 2. This is clearly useful information that readers will want to hear if they are considering

forming a writing group. To answer your questions, all the participants could work on posts at any time. Once they were ready, then they would be read at the group meetings and feedback would be given. This would usually take 20-30 minutes per article. The chairperson would be in charge of guiding the discussion, trying to avoid a "free-for-all" discussion.

Proposed action: We will include information about meeting length and size in section 2 where the writing process is described. We will also describe in more detail the responsibility of the group leader to guide the discussion so that it does not become a chaotic free-for-all.

Action taken: We have included the following text in section 2 (linked to the writing process figure):

*"Once the snack is published the writing process can start again, resulting in a continuous process and hence continuous improvement. The process is also flexible, and group leaders have the freedom to adapt it as they see fit. Most group meetings lasted between 1-2 hours once every 3-4 weeks. Each completed snack was discussed for 20-30 minutes depending on how many snacks were under consideration, and at what stage they were at. Some time was also often left at the end of the group meetings for general discussions and brainstorming. Some group meetings had up to 20 participants, but usually 5-10 people attended the meeting in the groups that regularly convened. The group leader was in charge of guiding the discussion and following the framework illustrated in Figure 2."*

**Reviewer quote: Do the groups use any online co-editing software (I'm thinking of something like Google docs) to share comments or are they all hand-written on printed copies?**

Reply: Again, this is useful information that we should include this information in the manuscript. Initially, we encouraged the participants to provide hand written feedback. One of the main reasons for this is that it encourages people to attend the meeting s and physically hand over the annotated document and explain why they made the changes. Editing software means that people can contribute remotely and might not turn up. However, online editing would be an excellent resource if virtual writing groups could be developed in this project, which is something we have considered before but not got funding for.

Proposed action: We'll explain in greater detail how the feedback is given, probably in section 2.

Action taken: We have included the following text in section 2 in connection with the writing process under "5. Feedback (Meeting)":

*"Before the meeting, the other group members should have read the author's snack and noted down feedback directly on to the snack itself. These notes would be passed to the author at the end of the meeting. We found that this encouraged meeting attendance. However, groups could also consider other feedback methods, like online editing software (e.g. Google docs). At the meeting....."*

**Reviewer quote:** On page 3 line 21, you seem to imply that improving basic writing skills will automatically translate into improved scientific writing skills. I think the former is necessary, but not sufficient, for the latter. Can you be a bit more explicit about some of the skills you do/do not think are covered by this process? I'm also thinking about how skills related to creating effective blogposts do/do not relate to other types of writing required by scientists (e.g. see next point).

**Reply:** We agree that the "former is necessary, but not sufficient, for the latter", however the form is *necessary,* and that's where ClimateSnack positions itself. To improve science writing and outreach, we must improve our basic writing skills.

This comment also inspired a healthy discussion amongst the co-authors. We feel that that many of the skills needed for quality blogging can be transferred to scientific writing. These are:
-basic writing skills
-critical thinking
-ability to summarize (conciseness)
-story-telling skills
-why it matters
-argument structuring

In the same note, we understand that the technical ability and understanding required for quality scientific writing cannot be gained from blogging experience.

Proposed action: We can certainly add a sentence where we clarify our position that blog-writing skills can improve scientific writing, but do not qualify an author to write quality scientific articles.

Action taken: We changed one of the later paragraphs in the Introduction to now read:

*"The objective of the ClimateSnack project was to encourage ECS's to self-organize writing groups to improve their basic writing skills, and thereby also their scientific writing skills. These basic writing skills alone are not sufficient to write quality scientific articles. However, these skills are important ingredients for overall improvement."*

**Reviewer quote:** This is more about the concept than the paper, but have you thought about using these groups to provide peer-review for other types of writing ECS's are faced with? I'm thinking about things like grant proposals, scientific papers, abstracts for conferences, etc. This might entice ECS's who aren't committed to writing blogposts, but would engage in activities more focused on something they already have to do.

**Reply:** It's really nice to read these suggestions and that our paper has made the reviewer think laterally like this.

These types of initiatives are not part of the concept directly. We have thought about things like this before, but we wanted to keep ClimateSnack as focused as possible.

However, that is not to say that these types of discussions have not occurred outside the groups or even within some groups on occasion.

Several of the co-authors commented on this issue. Since the writing groups create friendship, the members of one group asked each other for opinions and comments on other types of writing (mostly abstracts for conferences and travel grant applications).

In another group, a Ph.D. student, who participated in several of the meetings, had serious difficulties with writing in English. This was a particularly pressing concern for him as the student was nearing the end of their Ph. D. and needed to write up some of the results in a paper. For one of the Snacks the student brought along the abstract and a short section of a paper and received feedback much like a normal "snack". The student did not upload this to the website because it was material that would late be published in a peer reviewed journal. The student found the process very helpful, and seemed to take on board most of the copious feedback.

The challenge with other forms of writing is that they are often much longer than shorter blog posts. Several meeting would probably be needed to give constructive feedback on a single paper for example. Also conference abstracts should not be published online until later. This defeats the objective of the website, that we feel is an integral component of the ClimateSnack process.

Proposed action: Since other forms of writing are not a direct part of ClimateSnack, we would like to refrain from referring to them earlier in the paper. However, we will add a sentence or two explaining that the friendships and community built up around the writing groups allows us to seek out advice about other forms of writing. The reviewer could let us know if she would like us to include more details about specific examples that we have mentioned above.

Action taken: We have added the following text to the Discussion and Conclusion section to illuminate how the ClimateSnack community could also help participants with other types of writing:

*"No matter how long the different writing groups lasted, they brought ECS's together and created a forum for discussing writing skills for various goals such as grant proposals, scientific articles and conference abstracts. Friendships and community have been built around ClimateSnack (the present author group being an example) allowing ECS's to seek out advice and feedback even if their groups discontinued."*

**Reviewer quote: Along the same lines, for groups not comfortable with how to give feedback, I wonder if a structured rubric would be a good complement to the process depicted in Figure 2? I've been using rubrics based on the goals of the writing product in my undergraduate scientific writing class and it seems to help the students get started on first assessing the content (function) and then figuring out how the structure (form) could best support the ideas.**

Reply: The reviewer touches on an important point here. Confidence in writing is often reflected in confidence to give feedback. Rubrics are absolutely a valuable tool that we should certainly promote more actively and we have tried to promote via expert posts

on the website. However, we wanted to describe the process that we promoted from the beginning of the project. Therefore rubrics are not mentioned specifically in the article.

Proposed action:  We appreciate the reviewer's suggestion. If the reviewer agrees, we could a sentence or two in the section 2 about rubrics to inform the readership. However, we will have to mention that we have yet to actively use these in our groups.

Action taken: Although we completely agree that rubrics offer a powerful feedback tool, we decided not to mention them here. We want to concentrate on what we actually did in the ClimateSnack project. Having said this, we will promote the use of rubrics to all new groups in the future.

**Reviewer quote: Page 2, Line 11: I think a word is missing between "communicate" and "disciplinary"**

Proposed action:  Thank you for noticing. We will fix this.

Action taken: Text corrected.

**Reviewer quote: There seems to be inconsistency in whether or not the first line of a new paragraph is indented (e.g. page 6 line 22).**

Proposed action:  Thank you for noticing. We will fix this according to the guidelines for the EGU journals.

Action taken: Text corrected.

EXTRA ACTION TAKEN: We have made some small changes to sentences and phrases that we feel improve the readability. These can be seen in the file where we track changes.

**Response to Reviewer 2 RC2 and Editor EC1**
**hess-2016-13**
**Improving together: better science writing through peer learning**

**Reviewer quote:** **The paper is a descriptive account of the development of academic writing groups aimed at improving basic scientific writing skills for early career scientists working in climate science / geoscience… I certainly found myself trying to imagine setting something like this…**

**Reply:** This was nice to hear. We appreciate the reviewer's comment.

**Reviewer quote:** **Indeed, one is the actual origins of the ClimateSnack project itself, which is introduced rather unobtrusively *(line 20, page 3)* but I think a sentence or two about its background context would help the reader.**

**Reply:** We can certainly do this, however it will be a personal account, where the lead author will explain where the idea came from. He completed an online writing course (the exact one that we have available on the website) in 2011. During the course, a considerable online community developed; participants commented, peer-reviewed each others work and shared ideas via the course website. Once the course ended, this community disbanded, which was a shame. In order to keep the continuity going and to create a lasting community, the lead author decided that in-situ writing groups would be the perfect solution. The international community (which developed during the writing course) could be nurtured if lots of writing groups started, and if a website acted as a focal point.

Proposed action: Could the reviewer please indicate how much detail within this story he would like to have included in the paper?

Action taken: We added the following text to the introduction about the project's inspiration:

*"Online writing courses are often freely available and the development of the ClimateSnack project was inspired by one such course lead by Kristin Sainani at Stanford University (http://online.stanford.edu/course/writing-in-the-sciences). During the course, students peer-reviewed each other's writing and an active online discussion community flourished. However, when the course ended, the community dissolved. The founders of ClimateSnack wanted to duplicate this interactive forum and also encourage continuity. Therefore, in-situ writing groups were proposed. Discussion would then continue within the groups and between international participants via the project's website."*

The information about Kristin Sainani's course was added to the text above and therefore deleted from the paragraph before, where it was originally.

**Reviewer quote:** **Firstly, although the authors state that scientists '. . .must learn to better consider their audience, and communicate their science more clearly', for me, the paper wasn't especially clear on who precisely the ECRs are writing for.**

**The paper implies the audience is both scientists within their own discipline and those across disciplinary boundaries, so presumably the focus remains squarely on academic writing rather than drifting into popular science writing (for which there is a far richer science communication literature that is not called on here). I appreciate that there is a continuum of writing styles that can be invoked for each different audiences, but it would be helpful if the article could add a sentence making clear the specific readership that ClimateSnack participants are targeting, as that sets the rubric for all that follows in terms of how they prepare and hone their contributions.**

Reply: We fully agree that the definition of the audience is indeed very important and, as the reviewer pointed out, has an impact on the writing style. The definition of the audience in the reviewed submission is stated on page 4, lines 7+8:

*"The audience for these snacks are fellow ECS's. In other words, the audience is assumed to be scientifically literate but not from a single research discipline."*

Arguably, the target audience should be mentioned earlier in the introduction of the ClimateSnack project in the paragraph starting on page 3, line 20 for example.

We could also be clearer that we're not aiming at popular science. But that within the 'scientifically literate' sphere, there is still a wide spectrum of possible audiences with different levels of familiarity with the ideas and themes being discussed.

Proposed action: We will be happy to make changes as the reviewer suggests. We will move the information on the proposed audience to the Introduction and include more details as we indicated here.

Action taken: In the introduction, and deleted from section 2:

*"The audience for these snacks are fellow ECS's. In other words, the audience is assumed to be scientifically literate but not from a single research discipline. The participants therefore also gain experience in interdisciplinary communication, where audiences will have different levels of familiarity with the ideas and themes being discussed. Once an author has written an article, the rest of the group would then provide constructive feedback, which the author would use to improve the text."*

Reviewer quote: **Secondly, and in a similar vein, although the scheme seeks to improve 'basic writing skills, and thereby also their scientific writing skills' it is never made explicit what the deficiencies are that the initiative is trying to redress. As the paper notes, there are plenty of academic voices bemoaning the quality of academic writing but precious few that actually dissect the problem in a meaningful way; one telling exception is Goben, G.D. & Swan, J.A. 1990. The Science of Scientific Writing. American Scientist, 78 (Nov/Dec), 550-558. While the authors direct the reader to papers that presumably shed light on the substance of this problem, that is not especially helpful for an individual interested in improving their writing. Given that this paper attempts to set out the theoretical basis for this practice, it is important to be as explicit and transparent about how those championing the ClimateSnack initiative perceive the *fundamental***

**weaknesses and limitations in mainstream academic writing**. A short section or paragraph on this should be added

**Reply:** The reviewer touches on an extremely pertinent issue, and one that we should absolutely have made clearer!

So what are these "fundamental weaknesses and limitations in mainstream academic writing"? Despite us not mentioning them explicitly in the paper yet, we believe that these are addressed implicitly in the online courses that we supply plus the other expert advice. Gopen and Swan hit the nail on the head when they write (paraphrasing slightly):

*"Readers do not simply read; they interpret. (…) It has helped to produce a methodology based on the concept of reader expectations."*

Gopen and Swan also state: *"In our experience, the misplacement of old and new information turns out to be the No. 1 problem in American professional writing today."*

As Gopen and Swan suggest, these issues of reader-expectations and information-placement can be addressed by considering (and practicing) sentence structure, topic and stress positions, etc. In ClimateSnack, we feel that these deficiencies are common in both our scientific publications and other forms of general outreach. Hence we need to practice and improve. Blogging within a writing group environment is a powerful way to encourage this practice and improvement. We believe that all the writing skills we practice together are transferrable to scientific writing. But as we have responded to reviewer 1, we are very aware that scientific writing needs other writing skills too.

Proposed action: We will certainly include more detailed information in the Introduction about the "*fundamental weaknesses*" ClimateSnack attempts to address. We feel that this will also be made clearer when we include some screen shots as figures. See below.

Action taken: We have added the following sentences to the introduction based on the Gopen and Swan reference the reviewer mentioned:

*"Some of the fundamental weaknesses of scientific writing stem from issues of reader expectations and information placement (Gopen and Swan, 1990)"*

**Reviewer quote: Thirdly, the reason that the basic deficiencies need to be made more explicit is that it is not immediately clear how - or indeed, if - the ClimateSnack initiative is addressing the core communication issues raised by practitioners working in climate science arena. This is an arena that is pretty frequently addressed by those publishing in science communication. One prominent contributor is Richard Somerville, prof at Scripps and the science director of the nonprofit project 'Climate Communication', who has written and blogged extensively on this and highlights a range of issues that do not appear to feature in the ClimateSnack developmental process. For example, his review in Physics Today (Somerville, R.C.J. & Hassol, S.J. 2011. Communicating the science of climate change. Physics Today, October, 48-53.) critiques the conventional**

**academic model of writing and presents some clear recommendations for making climate science writing more accessible. It may be that the authors would disagree with his contentions, but the point is that it is impossible to tell because there is no indication of to what extent the now pretty extensive critical literature on climate science communication is infusing and informing the ClimateSnack initiative. To put it bluntly, is ClmateSnack simply a self-help support group for a particular scientific cohort or is it actively carrying forward the experience of climate science communicators?**

**If it is the former then OK but that more limited remit needs to be stated; if it is the latter then the paper needs to be far more explicit on how participants are building on what is out there.**

**Reply:** We agree this is another issue that we need to make clearer in the article. In particular, this speaks to the general objectives of the initiative that we need to expand upon.

Indeed Somerville dissects the climate communication problem. ClimateSnack is concerned with a small portion of the problem that Somerville refers to as: *why don't people believe climate science?* We are interested in his final point on this issue, namely: "*Not least important is how scientists communicate—or fail to do so. Reasons for that failure include what scientists talk about as well as how they talk about it. Narrative skills help reach people.*"

Somerville indicates that this last point resonates in the realm of science-to-public communication. But it is also a problem with science-to-science communication. Somerville excludes the point that communication also needs to improve *between* scientists. Scientists are people just like the public and need to be stimulated when they read in just the same way. This is especially so in this age of increased competition (amongst published articles) and increased interdisciplinarity. We don't/shouldn't write only for researchers in our own fields. We have to accept that anyone can search for and access our articles. We can have impact in any field of research, and this should influence our writing.

We certainly agree with many of Somerville's points. Indeed the "so what?" in Figure 3 should always come up in feedback discussions in ClimateSnack group meetings. However, we would argue that the elements should also be applied to our scientific writing as well. Indeed some style guides actively encourage the "so what?" to come very early in scientific articles (for example Joshua Schimel).

ClimateSnack is not "simply a self-help support group". Firstly, with all due respect, we would argue that using the word "simply" undermines the power and usefulness of support groups such as this within the research community. A self-help support group can provide enormous support for people to get the advice they need. Any research community is comprised of many people of non-english speaking backgrounds. The reviewer will hopefully agree that these types of non-judgemental support groups can give these researchers the platform they need to voice their concerns and ask for help. Secondly, we feel that ClimateSnack does actively carry forward the experience of science communicators in general, not solely climate science. This is achieved through the writing process we have suggested, in particular the funnel model, which was

developed by David Schultz. We stress in our paper that the discussions should be based on some prior knowledge. This knowledge is conveyed via the website in several different instances. We have an entire online writing course available, that Kristin Sainani has kindly allowed us to use. We also have expert advice columns and videos. Finally, we have a short book review section where some participants have written about books that helped them with their writing.

Proposed action: We need to clarify our position on the power of the "self-help group" in ClimateSnack. We also need to clarify how we are in fact promoting the advice of (climate) science communicators and experts through different media (books, videos, expert columns). All this information could be included under Section 2, which is concerned with the specifics of the writing process and the discussions.

Action taken: With respect to our stance on the "self-help group" issue, we added the following to the introduction:

*"These writing groups provide safe, non-judgemental environments for ECS's to practice, make mistakes and improve, which may be particularly pertinent for non-English speaking researchers."*

With respect to the writing issues we target, we have made this more explicit by editing/adding the following text to the section 2:

*"If this knowledge is difficult to come by, the ClimateSnack website freely provides an entire online writing course, advice from experts (see Figure 3), as well as book reviews from some of the ClimateSnack members. In this way, ClimateSnack targets the deficiencies in academic writing, and promotes the advice of experts. In addition, the discussion needs to be conducted with courtesy and humility."*

**Reviewer quote:** **Fourthly, could some indicative content from the site be included? It could be a screen- shot or two, or brief excerpts from posted articles. I found myself frustrated that all I was reading about what the process and could not view the product (at least, not without accessing the webpage - perhaps a deliberate ploy. I appreciate that it is a sensitive issue, but there must be examples of 'good practice' that the team feel show- cases what the ClimateSnack initiative can achieve in recasting academic writing. )**

**Reply:** We thank the reviewer for this suggestion. Indeed, a screenshot (example below) from the website should be included in the corrected manuscript. This is especially useful to present some titles as teasers of the products of the ClimateSnack community (which is now called SciSnack, as it has been expanded to welcome ECS's from all disciplines).

We feel that this important suggestion may also address some of the issues the reviewer has brought up earlier. Via this figure, we can show that ClimateSnack does try to carry *"forward the experience of […] science communicators"* as the reviewer mentioned earlier.

Proposed action: If we show four small screen shots, then we can include the homepage, a sample article, one of the video course pages and expert advice. These screen shots and accompanying text will help illustrate that we use the web site as an outward

broadcast tool, but also as a learning resource where we provide the knowledge required for informed feedback discussion and further learning.

[Figure]

*Example of the extra figure that could be included. (Top left) Homepage. (Top right) short excerpt from one of the articles. This one was also published in Norwegian in one of Norway's national newspapers. (Bottom left) One of the chapters in the video writing course: Cutting the Clutter with Kristin Sainani from Stanford University. (Bottom right) An excerpt from one of the expert advice columns by Dallas Murphy.*

Action taken: We have included the figure (with updated figure text) that we suggested here. The figure is mentioned in relation to the writing process in section 2. This includes the text from the above comment, and the following after "6. Improve and publish online":

*"Figure 3 includes some screenshots from the website, including the homepage and an example snack."*

Reviewer quote: **Finally, many readers of this paper will lament the omission of some kind of empirical analysis of its efficacy. The informal 'survey' of why groups succeed or not simply adds to the frustration of not getting a better sense of how effective this novel approach is; to make a useful contribution more specifics on what this survey involved should be given. Overall, I'm sympathetic to the nascent nature of the initiative and also to the difficulties in determining meaningful metrics, but I agree with the other reviewer that there are indicative measures that the authors could and should consider regarding readership and impact.**

**Reply:** It is understandable that both reviewers brought up this poignant issue. We have answered Reviewer 1 in detail on this issue. We paste this answer here for the present reviewer to consider:

Thanks to the reviewer for a very constructive suggestion. Since we received both the reviews we have had an intensive discussion within the author group about such metrics.

The author group agrees that getting formal metrics in retrospect would not be desirable. The metrics from the UEA group are very clearly described as "informal" and we only use these as indications of the effects.

Metrics are something the project managers should absolutely have considered at the beginning of the project. However, ClimateSnack has always been a voluntary project where many of us have used our free time, with little or no funding, to develop groups, support authors, and write ourselves. We feel that formal metrics would have taken considerable time to develop and instigate. This would have required considerably more funding.

The reason we think that this would have been more complicated than maybe first imagined, is that the effects of such writing groups are so multi-faceted. As we have discussed in the paper, it's not just about writing quality; the effects are also concerned with general confidence, critical thinking, and network building. We must also consider the writing *process* in addition to the quality of the final product.

We also discussed how we could have measured improvement in writing quality. This would likely have been left up to the participant to judge himself. One of our co-authors pointed out a substantial challenge with this. He told us that he was a very confident writer before he joined ClimateSnack. However, through the writing process and group feedback, he started to understand that his writing was not as skillful as he first assumed. If he had filled out a self-assessment form before and after his participation, he may have actually perceived a decrease in writing quality, whereas objectively his writing had actually improved.

Moreover, ClimateSnack is an initiative where virtually all participants are early-career researchers. Most objective metrics would require members to have relatively long control periods both before and after joining ClimateSnack. The former requirement already excludes the large majority of members, who joined ClimateSnack during their Ph.D.

As part of the review process we carried out a survey to gather information such as acceptance rates of paper and abstracts, success in applying to travel awards etc. However, we quickly realised that most of our members joined ClimateSnack very early during the career, and that the changes in the metrics perhaps reflect more the natural development of their scientific abilities than the benefits of our writing groups.

Our most important point is that we feel our whole paper is already a metric. Indeed, it is not a quantitative metric (as alluded to by the reviewers), but it is a narrative metric. We feel that this is both more valuable and robust than an *ex post* survey, which would

encounter all of the issues described above. The whole paper is built upon the narratives of 13 of the most active ClimateSnack members and others. Everyone in the author group has been a member of a ClimateSnack writing group. Some started groups that succeeded, whilst others started groups that dissolved. All the authors have built a network internationally (case in point, the present paper), and also extended their networks where they work.

Proposed action: We will add text explaining that we take a narrative approach in this paper and emphasizing how much the authors have contributed to this project.  We will also add text to say that the lack of quantifiable metrics may be a limitation, but that this is something we could consider for the future. Similar projects should certainly consider metrics from the beginning.

We hope this sounds reasonable to the reviewers. Unfortunately we never had funding or time to carry out a formal evaluation from the beginning of the project. Hopefully one day we will have the funding to develop these ideas further.

Action taken: We had to change around the end of the introduction slightly to accommodate these changes. The last paragraph of the introduction now reads:

"*In the next section, we introduce the learning process around which ClimateSnack was built. The successes and challenges that the different groups encountered will be presented in section 3. Results from an informal questionnaire answered by one group are presented to illustrate how the participants benefitted. Besides this, we did not evaluate ClimateSnack's impact using quantitative metrics from the beginning. This paper therefore takes a narrative approach and reports on the experiences of the members and their groups. These narratives are provided by the present paper's authors, all of whom have been involved in the project development. Some authors founded and managed ClimateSnack, whilst other authors started groups that are either still running or have since dissolved. All the authors have been active participants and have written and posted articles on the ClimateSnack website. We began to collate our experiences in a group meeting at the European Geosciences Union General Assembly in Vienna in 2015. Further discussion has taken place via email and video conferencing. In section 4, we generalize the lessons learnt and conclude with some thoughts about the future.*"

We also included the following in the conclusion, in order to explain the possible limitation of the lack of quantitative evaluation metrics:

"*Even though we did not evaluate the project using continuous quantitative metrics, this paper is based on the honest accounts and narratives of many of the ClimateSnack participants who make up the paper's author group. We would recommend that similar projects consider tracking evaluation metrics from the beginning. Through our collected experiences we have seen that ClimateSnack faced several challenges, but the successes show that peer learning through writing groups can be a valuable approach to achieve improved writing in science.*"

**Reviewer quote:** **On a related note, perhaps the authors could mention something more about the international community that has been fostered as a result of ClimateSnack?**

**Reply:** This was indeed one of our main aims in ClimateSnack. As we wrote in the Introduction:

"ClimateSnack has two unique elements: it is self-organized…. And it tries to build an international community…."

This international community was not achieved in the way we first perceived via the commentary and interaction on the website. However, the author group of the present paper exemplifies the community that did arise. ClimateSnack has been successful in bringing ECS's together. We have also arranged international workshops (separate from the writing groups), town hall events and seminars that have brought ECS's together.

**Proposed action:** We will certainly include information about the failure to build a community in the way we first envisaged. This should have already been mentioned in the discussion/conclusion section, since we stated it as one of the unique elements.

We can further refer to the present paper and organized events as examples of the networks ClimateSnack has motivated. After all, with all our (the co-authors) varied scientific backgrounds, it is unlikely we would have written a paper together if it weren't for our shared experiences through ClimateSnack.

**Action taken:** Both the international community and the community are now mentioned in the last paragraph with the following sentences:

*"It is true that the international network did not develop in the way the ClimateSnack founders first envisaged. However, within successful writing groups, solid support networks arose and many members discovered that writing could be a pleasurable activity."*

**Reviewer quote: In summary, the paper is an enthusiastic but rather uncritical account of one initiative to counter perceived limitations in our current academic writing provision. While I share many of the authors' basic contentions and find the ClimateSnack an intriguing and welcome development, the paper as it stands lacks substance in key areas and I would ask the authors to attempt to address these in their revisions.**

**Reply:** As we mentioned above, we will include clearer indications of the communication problems that ClimateSnack attempts to tackle. We think that this gives a more holistic picture of the project.

We would however argue that we have in fact provided a rather critical account of the ClimateSnack project. We have fully disclosed the limitations and failures we encountered setting up groups. We have described writing groups that have both succeeded and dissolved. We have further stated that the majority of groups dissolved. We have also been open about the project aims that were not achieved. For example we admit that we were not clear enough about the projects objectives to start with, and that this may have caused confusion amongst new groups. As stated above, we will also add more text about the failure to nurture the international community in the way we first envisaged.

**Proposed action:** If the reviewer still thinks we are being uncritical, then we would like to hear some specific suggestions to address this point. It is important for us to come

across as transparent and honest in our accounts of ClimateSnack's performance. Even though we provide few quantitative metrics, we want our narratives to be as relevant, clear and informative as possible.

Thank you for your honest and constructive suggestions and criticism!

Action taken: No extra action was taken here, since we still stand by our claim that we have in fact provided a rather critical account of the ClimateSnack project with open discussions about groups that both continued and disbanded.

EXTRA ACTION TAKEN: We have made some small changes to sentences and phrases that we feel improve the readability. These can be seen in the file where we track changes.

**Improving together: better science writing through peer learning**

M. A. Stiller-Reeve[1], C. Heuzé[2,3], W. T. Ball[4], R. H. White[5], G. Messori[6], K. van der Wiel[3], I. Medhaug[7], A. Eckes[3], A. O'Callaghan[3], M. J. Newland[3], S. Williams[8], M. Kasoar[8], H. E. Wittmeier[9], V. Kumer[10]

[1] Uni Research Climate, Bjerknes Centre for Climate Research, Bergen, Norway
[2] Department of Marine Sciences, University of Gothenburg, Gothenburg, Sweden
[3] Centre for Ocean and Atmospheric Sciences, School of Environmental Sciences, University of East Anglia, Norwich, UK
[4] PMOD/WRC, Dorfstrasse 33, 7260 Davos Dorf, Switzerland
[5] Joint Institute for the Study of the Atmosphere and Ocean, University of Washington, Seattle, USA
[6] Department of Meteorology, Stockholm University and Bolin Centre for Climate Research, Stockholm University, Stockholm, Sweden
[7] Institute for Atmospheric and Climate Science, Department of Environmental System Science, ETH Zurich, 8092 Zurich, Switzerland
[8] Department of Physics and Grantham Institute, Imperial College London, London, SW7 2AZ, United Kingdom
[9] Department of Earth Science, University of Bergen, Allégaten 41, N-5007 Bergen, Norway
[10] Geophysical Institute, University of Bergen, Bergen, Norway

*Correspondence to*: M. A. Stiller-Reeve (Mathew.reeve@uni.no)

**Abstract:** Science, in our case climate- and geo-science, is increasingly interdisciplinary. Scientists must therefore communicate across disciplinary boundaries. For this communication to be successful, scientists must write clearly and concisely, yet, the historically poor standard of scientific writing does not seem to be improving. Scientific writing must improve and the key to long-term improvement lies with the early-career scientist (ECS). Many interventions exist for an ECS to improve their writing, like style guides and courses. However, momentum is often difficult to maintain after these interventions are completed. Continuity is key to improving writing.

This paper introduces the ClimateSnack project, which aims to motive ECS's to develop and continue to improve their writing and communication skills. The project adopts a peer learning framework where ECS's voluntarily form writing groups at different institutes around the world. The group members learn, discuss and improve their writing skills together.

Several ClimateSnack writing groups have formed. This paper examines why some of the groups have flourished and others have dissolved. We identify the challenges involved in making a writing group successful and effective, notably the leadership of self-organized groups, and both individual and institutional time management. Within some of the groups, peer learning clearly offered a powerful tool to improve writing as well as bringing other benefits, including improved general communication skills and increased confidence.

Mathew 10/6/2016 13:35
**Slettet:** of scientific writing

Mathew 10/6/2016 10:28
**Slettet:** E

Mathew 10/6/2016 10:28
**Slettet:**

Mathew 10/6/2016 10:28
**Slettet:** C

Mathew 10/6/2016 10:28
**Slettet:** S

Mathew 10/6/2016 10:30
**Slettet:** -

Mathew 10/6/2016 13:36
**Slettet:** been

Céline Heuzé 9/6/2016 18:11
**Slettet:**

[revised manuscript text omitted]

Céline Heuzé 9/6/2016 18:14
**Slettet:** ,

Céline Heuzé 9/6/2016 18:14
**Slettet:**

Céline Heuzé 9/6/2016 18:15
**Slettet:** then

Céline Heuzé 9/6/2016 17:57
**Slettet:** e

Mathew 10/6/2016 13:42
**Slettet:** In the writing groups, ECS's could discuss and practisce writing in a safe, and supportive environment, and gain confidence as writers (Ferguson, 2009).

familiarity with the ideas and themes being discussed. Once an author had written an article, the rest of the group provides constructive feedback, which the author uses to improve the text. The author publishes the finished article on the project's website (www.climatesnack.com). The web page also acts as a space for the participants to network with like-minded researchers from around the world. ClimateSnack has two unique elements: it is self-organized by ECS's and it tries to build

5   an international community around writing skills in science.

In the next section, we introduce the learning process around which ClimateSnack was built. The successes and challenges that the different groups encountered will be presented in section 3. Results from an informal questionnaire answered by one group are presented to illustrate how the participants benefitted. Besides this, we did not evaluate ClimateSnack's impact using quantitative metrics from the beginning. This paper therefore takes a narrative approach and

[revised manuscript text omitted]

Céline Heuzé 9/6/2016 18:02

**Slettet:**

Céline Heuzé 9/6/2016 18:03

**Slettet:** c